# How information about perpetrators' nature and nurture influences assessments of their character, mental states, and deserved punishment

Julianna M. Lynch[1], Jonathan D. Lane[2], Colleen M. Berryessa[3], Joshua Rottman[1]*

**1** Department of Psychology, Franklin & Marshall College, Lancaster, Pennsylvania, United States of America, **2** Department of Psychology and Human Development, Vanderbilt University, Nashville, Tennessee, United States of America, **3** School of Criminal Justice, Center for Law and Justice, Rutgers University, Newark, New Jersey, United States of America

* jrottman@fandm.edu

**Data Availability Statement:** All data files have been uploaded to the Open Science Framework

## Abstract

Evidence of perpetrators' biological or situational circumstances has been increasingly brought to bear in courtrooms. Yet, research findings are mixed as to whether this information influences folk evaluations of perpetrators' dispositions, and subsequently, evaluations of their deserved punishments. Previous research has not clearly dissociated the effects of information about perpetrators' genetic endowment versus their environmental circumstances. Additionally, most research has focused exclusively on violations involving extreme physical harm, often using mock capital sentences cases as examples. To address these gaps in the literature, we employed a "switched-at-birth" paradigm to investigate whether positive or negative information about perpetrators' genetic or environmental backgrounds influence evaluations of a perpetrator's mental states, character, and deserved punishment. Across three studies, we varied whether the transgression involved direct harm, an impure act that caused no harm, or a case of moral luck. The results indicate that negative genetic and environmental backgrounds influenced participants' evaluations of perpetrators' intentions, free will, and character, but did not influence participants' punishment decisions. Overall, these results replicate and extend existing findings suggesting that perpetrators' supposed extenuating circumstances may not mitigate the punishment that others assign to them.

## Introduction

When individuals violate moral norms, others may consider multiple factors when deciding how much punishment the violators deserve. These judgments regarding persons' culpability for violations are sometimes addressed in the legal system and also arise in more common, everyday folk evaluations. Studies in moral psychology are often designed to identify factors that influence blame and punishment judgments, including the harm caused by the action, the

repository, at the following link: https://osf.io/34v29/.

**Funding:** The authors received no specific funding for this work.

**Competing interests:** The authors have declared that no competing interests exist.

rules that are broken, violators' intentionality, motivations, character, and psychological profiles [1–6]. This previous work reveals that people focus primarily on the severity of outcomes when deciding how much to punish a perpetrator [7–8]. However, demographic and dispositional qualities of the perpetrator (e.g., socioeconomic status; character traits) may mitigate the amount of punishment that third parties assign, both in legal contexts [9–12] and in everyday moral judgments [5].

As neuroscience advances and its applications become more widespread, research has begun to investigate how neuroscientific evidence influences people's judgments of criminal behavior [13–17]. This research is often described as evaluating the role of biological factors in folk evaluations of moral culpability. However, this interpretation is limited given that neurological, psychological, and behavioral phenomena are products of complex, dynamic interactions between genetic substrates and environmental factors, such as socio-economic status, parenting, and educational access [18–20]. It is currently unclear whether the general public appreciates this complex interaction between genes and environment; for example, some people might inaccurately interpret neurological evidence as reflecting nature alone while others might assume that nurture plays a large role. In the current studies, we systematically tease apart biological and environmental factors in order to examine how individuals simultaneously weigh explicit information about violators' nature and nurture when evaluating the violators' mental states, character, and deserved punishment.

Previous research on the extent to which biological and environmental factors influence evaluations of transgressors has yielded mixed findings [21–23]. In cases in which neuroscientific and psychological evidence humanizes the offender by providing a rationale for his or her behavior—for example, impaired reasoning ability—judgments of culpability and punishment typically decrease [3, 19, 24–25]. For instance, mental disorders such as psychopathy, Autism Spectrum Disorder, and schizophrenia have been found to mitigate punishment because they provide insight into the mental state of the offender, and his or her impaired ability to reason, which reduces blame for wrongful actions [13, 19, 24]. Further, several studies have found that attributing behavior or characteristics to genetic influences can mitigate punishment assessments [26–29].

On the other hand, evidence that highlights genetic substrates of disorders can also emphasize the stability of an antisocial trait, in turn increasing evaluations of violators' dangerousness [13, 17, 25, 30] and inferences that violators are less likely to reform [3, 17, 21, 24]. Such evaluations of dangerousness, particularly related to genetics, can aggravate punishment assessments [5, 31–33]. When a genetic predisposition for criminality is paired with negative environmental experiences, such as abuse, longer prison sentences are often recommended, particularly related to fear of the offenders' dangerousness [34].

Here, to augment the existing literature, we contribute to the ongoing debate about how genetic and environmental factors influence assessments of perpetrators' blameworthiness by directly separating information about perpetrators' "nature" and their "nurture". In one of the few prior studies to explore this issue, participants received separate genetic and environmental information about a defendant and participants were asked to make capital punishment decisions [3]. Our study differs from this previous research in several fundamental ways. First, Gordon and Greene [3] presented information about a perpetrator in a structure which mimicked a legal case, involving testimony from a forensic psychiatrist and an expert medical geneticist. Respondents were also pre-screened for their eligibility to participate in a real capital sentence case. Although these features increase the ecological validity of findings with regard to that specific circumstance (capital sentence decisions), it is not clear whether the findings would generalize to other populations, violations, and punishments. Our paradigm is more expansive, including a rich variety of more common wrongdoings: a simple harm

violation, an impure act, and a case of moral luck. It is critical to study everyday transgressions, as previous literature hints that genetic explanations may be less relevant for punishment decisions in these situations as compared to criminal cases [34]. We measured the amount of punishment deserved on a continuum (as opposed to asking for dichotomous decisions, like those faced for capital sentence cases), which allowed us to capture greater variability in punishment ratings. Additionally, we isolated the specific genetic and environmental factors without including other dispositional, demographic, or mental health factors that could confound this information.

The current studies utilize a "switched-at-birth" task [35–36] in order to clearly test how evidence about perpetrators' innate factors and contextual factors influence third parties' evaluations of blame and punishment. In this paradigm, participants are presented with a vignette in which a set of parents have a baby who accidentally gets switched at the hospital and is raised by a different set of parents. Participants are then asked whether the child will grow up to have characteristics like its birth parents or like the new parents. The current studies examine adults' judgments, but we specifically created scenarios that could also eventually be used to examine developmental patterns in such judgments. Thus, scenarios in this study were worded such that they would be appropriate for children and for adults with a variety of educational backgrounds. To these ends, the switched-at-birth scenarios and questions employed relatively simple language and the perpetrator was described as being a child (this was specified as a seven-year-old child in Studies 2 and 3). Seven-year-olds are typically considered to have reached the "age of reason" in which they are judged as mature moral agents [37]. Thus, the vignettes seem appropriate for both adults and children, and the simplification of scientific terminology also provides accessibility for wider ranges of audiences who may not have a deep understanding of the nature of genetics or environmental factors. Overall, these aspects of the study design allowed us to precisely test for the effects of explicit information about nature, nurture, and their potential interaction, in a way more generalizable to folk accounts of blameworthiness and deserved punishment.

Although the actual origins of antisocial behavior appear to be influenced by gene-environment interactions that are still not fully understood [38], we aimed to examine people's intuitions about person-centered factors that are of either nature- or nurture-based origin, as people often tend to think of these as separate and discrete causes [39]. Assessments of actions as either being caused by nature or nurture may influence the assumed degree of control a perpetrator is thought to have as over his or her actions, the goodness/dangerousness of the perpetrator, and the degree and type of punishment the perpetrator should receive for his/her actions. In our preregistration (https://osf.io/34v29/), we hypothesized that information about a perpetrator's bad genes and information about their bad environment would independently mitigate judgments of their character (e.g., dangerousness), free will, blame, and deserved punishment. In light of previous evidence suggesting that extenuating circumstances do not heavily impact blame, another plausible hypothesis is that participants will focus primarily on the harmful consequences, rather than the protagonist's nature or nurture when evaluating transgressors, and will not take genetic or environmental factors into account. As blame is often heavily contingent on negative outcomes [7], Studies 2 and 3 explored the extent to which information about a perpetrator's genetic predispositions and environmental factors impact evaluations of the perpetrator in the absence of causing harmful outcomes—specifically, in cases of violating purity norms (Study 2) and in cases of moral luck (Study 3). All studies were approved by the Franklin & Marshall College Institutional Review Board, Protocol #R_2TTZdYDcDkHS4Nd. We have reported all measures, conditions, and data exclusions.

## Study 1

### Methods

**Participants.** Participants were recruited through Amazon Mechanical Turk and paid $0.65 each. They were restricted to workers in the United States with a minimum HIT approval rate of 95%. The survey was completed by 201 participants, but 34 (17%) were excluded for providing one or more inaccurate answers on comprehension check questions (described below), leaving us with data from 167 participants (94 female, $M_{age}$ = 37.65, $SD_{age}$ = 10.75). A power analysis using G*Power 3.1 [40] indicated that we required a minimum of 158 participants to detect medium-sized main effects (e.g., $fs \geq .25$) with statistical power at the recommended .80 level and $\alpha$ = .05 [41]. With 167 participants, our actual power was .825.

**Design and procedure.** Participants completed the survey on Qualtrics. After providing consent and identifying their gender, each participant was randomly assigned to read a single gender-matched scenario about an individual who was switched-at-birth (and thus raised by biologically unrelated parents) and who later committed a moral transgression. Specifically, vignettes described a story in which a baby (Jane or Fred) was born to parents who had either good, bad, or neutral genetic predispositions, but was accidentally switched at the time of birth and thus grew up with different parents in either a good, bad, or neutral environment. The first part of the story described the background of the birth parents (the "nature" component). Specifically, the story described the birth parents as having "something in their blood and brain that make them very [nice, mean, ordinary]." (We chose to use this language, rather than language about genes or DNA, in order to make the vignette broadly accessible to a range of audiences by describing proximate, familiar biological causes rather than distal, less familiar biological causes.) The second part of the story was the transition, which remained the same across all conditions and described Jane/Fred being switch-at-birth to biologically unrelated parents. Next, the living conditions of the unrelated parents were described (the "nurture" component). Lastly, participants received information that Jane/Fred had been caught throwing rocks at other children on the playground (the transgression). The transgression remained the same across all conditions. Thus, participants were randomly assigned to one of nine conditions resulting from a 3 (Genes: Good, Bad, or Neutral) X 3 (Environment: Good, Bad, or Neutral) design. Using the switched-at-birth design allowed us to juxtapose information about a perpetrator's genetic versus environmental circumstances (see Fig 1 for details of the study design).

The impact of a perpetrator's good, neutral, or bad genes and good, neutral, or bad rearing environment were explored for seven dependent variables, which participants responded to using 4-point Likert scales. First, participants were asked questions about the perpetrator's intention to commit the transgression and the perpetrator's decision-making ability. Questions 1 and 2 asked about the perpetrator's mental states, specifically *Intention*: "Deep in her heart, did Jane/Fred want to hurt the other children?" and *Free Will*: "Could Jane/Fred have decided to be nice to the other boys and girls instead?" The choice options for Intention and Free Will were: Definitely not, Maybe not, Maybe yes, and Definitely yes. Next, Questions 3–5 asked about the *Punishment* that the perpetrator deserved: "How much trouble should Jane/Fred get in for hurting the other children?"; "Do you think Jane/Fred should be put in time-out?"; and "Do you think Jane/Fred should get their toys taken away?" The choice options for Time-Out and Toys Confiscated were: Definitely not, Maybe not, Maybe yes, and Definitely yes, and the choice options for Trouble were: None at all, A little bit, A lot, and Really a lot. Lastly, Questions 6 and 7 asked about the perpetrator's *Character* traits: "How dangerous is Jane/Fred?" (choice options were: Not dangerous at all, A little dangerous, Pretty dangerous, and Very dangerous) and "How good is Jane/Fred?" (choice options were Not good at all, A

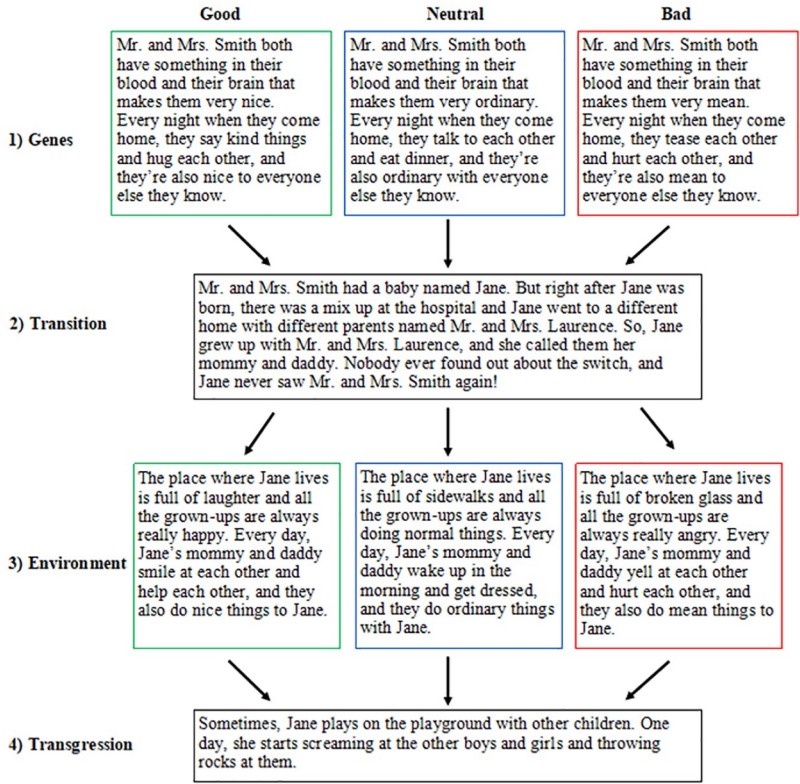

**Fig 1. Flow chart of Study 1 switched-at-birth vignettes and condition assignment.** Scenarios with good, bad, or neutral genes were randomly paired with good, bad, or neutral environments for a total of nine possible conditions. All participants received the same transition and transgression.

little good, Pretty good, and Very good). These were followed by comprehension questions that asked: "What happened to Jane/Fred right after she was born?" (open-ended response); "What do you think about Mr. and Mrs. Smith? Are they really nice, sort of nice, sort of mean, or really mean?"; "What do you think about the place where Jane/Fred lives? Is it really good, sort of good, sort of bad, or really bad?" Incorrect answers on any of these three comprehension questions led to exclusion.

We attempted to form three composites from these seven variables: Mental States, Character, and Punishment, in line with a preregistered decision (https://bit.ly/2WHWb7X). However, questions about Intention and Free Will did not form a reliable composite involving Mental States ($\alpha = .292$), so these questions were kept separate in analyses. (This was also true in Study 2, $\alpha = .299$, and in Study 3, $\alpha = -.291$). The questions assessing Dangerousness (reverse coded) and Goodness formed a reasonably reliable composite measure ($\alpha = .738$), so these questions were combined (averaged) into a Character variable. Similarly, the Trouble, Timeout, and Toys Confiscated questions formed a reasonably reliable composite measure ($\alpha = .676$), so these questions were combined (averaged) into a Punishment variable.

In this study and in Studies 2 and 3, participants were also asked to explain each of their ratings. These open-ended responses indicated that we successfully implied a deep, heritable biological characteristic in our vignettes. In Study 1, 15% of participants (14% in Study 2 and 9% in Study 3) spontaneously mentioned "genes", "genetic", "inherited", or "DNA" when explaining their closed-ended evaluations. This frequency of unprompted references to genetic inheritance (approximately twice the frequency of references to "blood" or "brain") suggests that

participants generally interpreted information about the target's blood and brain as being indicative of an underlying genetic predisposition, as was expected. Other details about these open-ended responses are beyond the scope of this paper, and are not discussed further.

Finally, because previous work has indicated that essentialist beliefs could influence the perceived relevance of a person's genetic background for evaluating their blameworthiness [3, 13, 17, 19, 24, 31, 42], we administered questionnaires to assess individual differences in overall essentialist tendencies at the end of this study and the two subsequent studies. In Study 1, we combined validated scales measuring overall genetic essentialism [43] and social essentialism [44]. Study 2 incorporated a measure of overall genetic essentialism [43], and Study 3 included additional measures of discreteness (i.e., distinctness or uniqueness), informativeness (i.e., revealing or instructive indications of personal dispositions), and biological basis [45]. In all studies, these measures were presented after participants had answered all questions about the focal scenarios. These measures did not achieve internal consistency in the first study, and did not yield informative findings in Studies 2 or 3, so we do not discuss them further (see the S1 File for the scales that were used; results are available from the authors upon request).

## Results

Four two-way Analyses of Variance (ANOVAs) were conducted to evaluate the effects of information about the perpetrator's Genes (3: Good, Bad, or Neutral) and the perpetrator's Environment (3: Good, Bad, or Neutral) on participants' evaluations of the perpetrator's intention, free will, character, and deserved punishment (see Fig 2). These analyses revealed a main effect of Genes on evaluations of Free Will, $F(2, 158) = 6.852$, $p = .001$, $\eta^2_p = .080$, which were lower when the perpetrator possessed bad genes as opposed to neutral genes, $p = .008$, or good genes, $p = .002$ (these and all subsequent post-hoc tests are corrected using Tukey's HSD). There was also a main effect of Genes on evaluations of the perpetrator's Intention, $F(2, 158) = 3.404$, $p = .036$, $\eta^2_p = .041$, which were elevated when the perpetrator possessed bad genes versus good genes, $p = .038$. Finally, there was a main effect of Genes on evaluations of Character, $F(2, 158) = 5.029$, $p = .008$, $\eta^2_p = .060$, which were more positive when perpetrators possessed good genes as opposed to bad genes, $p = .010$, and marginally more positive when they had good genes as opposed to neutral genes, $p = .065$. There was no effect of Genes on Punishment ratings, $F(2, 158) = .524$, $p = .593$, $\eta^2_p = .007$. Overall, information about the quality of perpetrators' genes influenced evaluations of their free will, intention, and character, but did not influence evaluations of how much punishment they deserved for their transgression.

These ANOVAs also uncovered an effect of Environment on evaluations of Character, $F(2, 158) = 4.909$, $p = .009$, $\eta^2_p = .059$, driven by participants' evaluations of the perpetrator as having a more positive disposition if they had been raised in a neutral versus bad environment, $p = .005$. There were no main effects of Environment on any other dependent variable; Intention: $F(2, 158) = .049$, $p = .952$, $\eta^2_p = .001$; Free Will: $F(2, 158) = .985$, $p = .376$, $\eta^2_p = .012$; Punishment: $F(2, 158) = 1.117$, $p = .330$, $\eta^2_p = .014$.

The sole interaction of information about the perpetrator's Genes and Environment was found for evaluations of their Free Will, $F(4, 158) = 2.496$, $p = .045$, $\eta^2_p = .059$. Thus, the main effect of information about genes on evaluations of free will (reported earlier) was moderated by information about the perpetrator's environment. When the perpetrator was raised in a bad environment, information about their genes was irrelevant. However, when the perpetrator was raised in a neutral or positive environment, they were judged to have greater free will if they had neutral genes (neutral environment: $p = .051$; good environment: $p = .002$) or good genes (neutral environment: $p = .001$; good environment: $p = .003$) as opposed to bad genes. There was no significant interaction between Genes and Environment for any other dependent

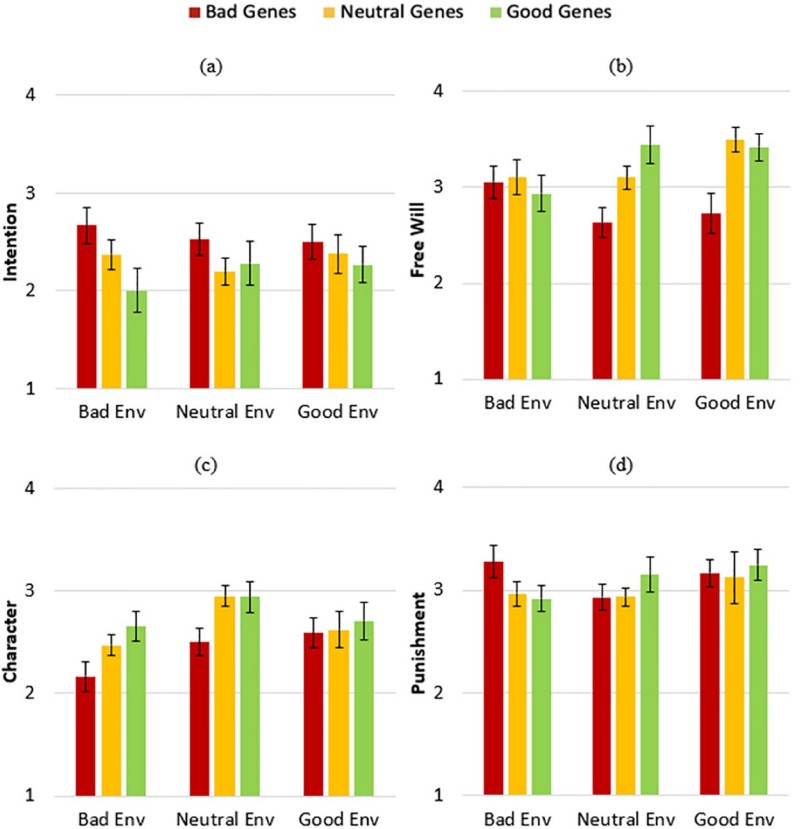

**Fig 2. Study 1 results.** Participants' average ratings (ranging from 1 to 4) of how much the perpetrator (a) desired to commit the transgression; (b) could have instead decided to act nicely; (c) had a positive character; (d) deserved punishment. Error bars represent standard errors of the mean.

measure; Intention: $F(4, 158) = .578$, $p = .679$, $\eta^2_p = .014$; Character: $F(4, 158) = .814$, $p = .518$, $\eta^2_p = .020$; Punishment: $F(4, 158) = .992$, $p = .414$, $\eta^2_p = .024$. Thus, while the quality of perpetrators' upbringing (i.e., environment) only directly influenced participants' evaluations of their character, it interacted with information about the perpetrator's genes to affect inferences about their free will. But this information about the perpetrator's upbringing showed no influence on inferences about the perpetrator's desire to hurt others or judgments of how much punishment they deserve.

## Discussion

Our results reveal that individuals account for the quality of a perpetrator's genetic background when inferring the perpetrator's intention to commit a transgression and their free will to commit a transgression (particularly in decent rearing environments), indicating that "bad genes" increase attributions of desire to commit immoral actions while reducing assumed controllability over these actions. This is consistent with previous research indicating that genetic information may affect inferences about persons' mental states, ability to reason, and personal jurisdiction over their actions [3, 13, 17, 19, 24–25]. Genetic background, in addition to the quality of a perpetrator's environment, was also taken into consideration when evaluating the perpetrator's character. The more negative the perpetrator's genetic background (i.e., the meaner the birth parents) and the more negative the environment in

which the perpetrator was raised, the more negatively participants evaluated the perpetrator's character. This is consistent with previous findings that genetic evidence conveying abnormal or negative information about an individual can sometimes increase impressions of dangerousness because the behavior is considered more persistent and enduring in nature [3, 17, 21, 24].

Yet, neither the quality of the perpetrator's genes nor the quality of their environment influenced the amount of punishment that individuals assigned to the perpetrator. Thus, qualities of the perpetrator's background influenced individuals' impression of the perpetrator's mental states and character, but had no bearing on their punishment decisions. This finding suggests that the crime itself (i.e., causing harm) may be more influential than the perpetrator's background when assigning punishment. Because the transgression (screaming and throwing rocks) was an egregious case of harm, participants may have focused wholly on the outcome of the action rather on the perpetrator's background when making punishment decisions. If so, then the influence of background information might be more apparent in cases of harmless transgressions, for example, for violations of moral purity. Thus, in Study 2, we compared the effects of background information for two types of violations: a harm violation and a purity violation.

## Study 2

Punishment appears to function as a result of a two-part system of blaming: (1) a person violates a moral code or social norm and (2) correspondingly is cognitively and socially blamed for the violation [4]. Society recognizes the wrongness of the violation and attempts to alter or reform the behavior by implementing punishing consequences, either socially or legally. Additionally, information about the person and events that precede the violation often influence judgments of wrongness and blameworthiness. For example, a person's intentions and the amount of harm caused by their actions are taken into account when determining how much blame and punishment they deserve [4, 7].

However, evaluations of purity transgressions have been found to operate differently from harm transgressions [46–48]. When considering strange or taboo behaviors (e.g., incest, cannibalism), people are more likely to provide person-centered explanations for why the actor committed the violation, meaning that they consider the perpetrator's character traits rather than the qualities of their action or its outcome [49]. Because judgments of purity violations involve a greater focus on the character of transgressors, it is possible that judgments of people who violate purity norms may rely more heavily on information about the perpetrators' genetic and environmental histories, relative to judgments of people who violate harm norms. In other words, even though the results from Study 1 indicated that genetic and environmental backgrounds are irrelevant for folk punishment decisions, this may differ for transgressions that do not have salient consequences or victims.

Similar to the previous study, Study 2 was an investigation of how genes and rearing environments influence evaluations of intention, free will, character, and punishment. However, in Study 2, we compared the harmful violation from Study 1 (throwing rocks) with a purity violation (public nudity/offensive language) to investigate if the type of transgression mattered when determining the influences of nature and nurture on punishment and other judgments. Because no direct harm was caused in the purity violation, and previous research has shown that character is a more relevant factor for evaluating impure acts compared to harmful acts [49–50], we expected information about the perpetrator's genetic and environmental background to play a larger role in participants' punishment decisions.

## Methods

**Participants.**   Participants were recruited through Amazon Mechanical Turk and paid $0.60 each. They were again restricted to workers in the United States with a minimum HIT approval rate of 95%. The survey was completed by 154 participants, but 14 (9%) were excluded for providing one or more inaccurate answers on comprehension check questions, leaving us with data from 140 participants (79 female, $M_{age}$ = 34.09, $SD_{age}$ = 10.12). A power analysis using G*Power [40] indicated that we required at least 128 participants to detect medium-sized main effects (e.g., $fs \geq .25$), with statistical power = .80 and $\alpha$ = .05. With 140 participants, our actual power was .836.

**Design and procedure.**   Participants completed the survey on Qualtrics. After providing consent and identifying their gender, each participant was randomly assigned to read a single gender-matched scenario in one of eight conditions, which varied by type of Genes (Good or Bad), type of Environment (Good or Bad), and type of Transgression (Harm, which involved throwing rocks and yelling, or Purity, which involved running around naked and shouting obscenities). Just as in Study 1, the switched-at-birth paradigm [35–36] was used, and this information was followed by a description of a single moral transgression. Unlike Study 1, no neutral conditions of genes or environments were included.

The harm transgression remained the same as in Study 1, with the exception of specifying the age of the perpetrator, saying that he/she is seven years old at the time of the transgression. The new purity violation was: "Jane is now seven years old. One day during recess, Jane decides to take off all her clothes and she starts running around the playground naked. She shouts dirty words while she runs" (see Fig 3). The participants were then asked seven different questions, each testing a different dependent variable (Intention, Free Will, Good-ness, Dangerousness, Punishment, Timeout, Toys Confiscated). The variables examined were the same as in Study 1, but the phrasing of the Questions 1–3 were changed slightly in the Purity condition to reflect the new transgression. Question 1 was: "Deep in her heart, did Jane/Fred want to run around the playground like this?". Question 2 was: "Could Jane/Fred have decided to keep her clothes on and play nicely with the others instead?". Question 3 was: "How much trouble should Jane/Fred get in for running around the playground like this?"

All questions were recorded on the same 4-point Likert scales used in Study 1 and compre-hension check questions were also included. One question was changed to provide more clar-ification: "Remember, Mrs. Smith is the person who got pregnant with Jane. What do you think about Mrs. Smith?" Responses were recorded using the same scale as in Study 1. As in Study 1, we created a *Character* composite which included questions about Dangerousness (reverse coded) and Goodness ($\alpha$ = .595); and a *Punishment* composite which included the Trouble, Timeout, and Toys Confiscated questions ($\alpha$ = .703). Although the reliability of the Character composite is modest, we retain it here given that the same composite is used (and has an $\alpha$ > .70) in Studies 1 and 3.

## Results

A series of four three-way ANOVAs tested for effects of Genes (Good vs. Bad), Environment (Good vs. Bad), and Transgression (Harm vs. Purity) on evaluations of intention, free will, character, and deserved punishment (see Fig 4). These analyses revealed a marginal effect of Genes on evaluations of the perpetrator's Character, $F(1, 132) = 3.513$, $p = .063$, $\eta^2_p = .026$, as evaluations of character were more positive when the perpetrator possessed good (vs. bad) genes. All other effects of Genes were non-significant; Intention: $F(1, 132) = .232$, $p = .631$, $\eta^2_p = .002$; Free Will: $F(1, 132) = 1.715$, $p = .193$, $\eta^2_p = .013$; Punishment: $F(1, 132) = .147$,

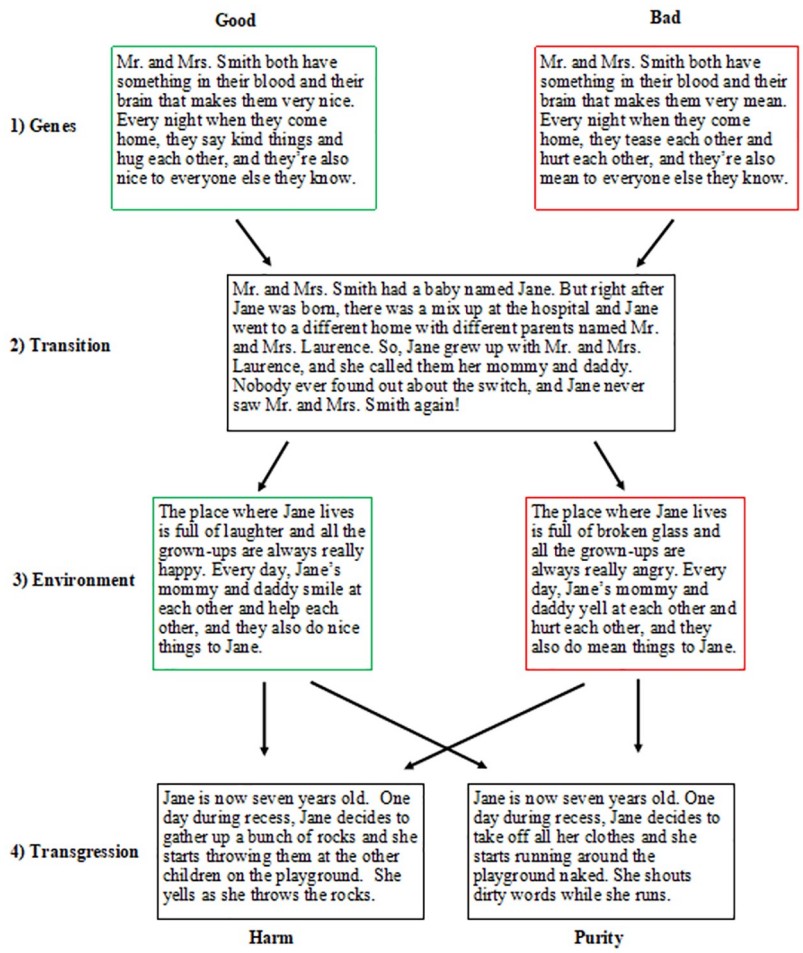

**Fig 3. Flow chart of Study 2 switched-at-birth vignettes and condition assignment.** Scenarios with good or bad genes were randomly paired with either good or bad environments and then randomly assigned to either the harm or purity violation, resulting in a total of eight possible conditions. Every participant received the same transition.

$p = .702$, $\eta^2_p = .001$. Thus, the perpetrator's possession of good or bad genes slightly influenced evaluations of character, but did not influence any other morally relevant assessments.

There were main effects of Environment on evaluations of Intention, $F(1, 132) = 3.939$, $p = .049$, $\eta^2_p = .029$; Free Will, $F(1, 132) = 4.970$, $p = .027$, $\eta^2_p = .036$; and Character, $F(1, 132) = 7.326$, $p = .008$, $\eta^2_p = .053$. When a perpetrator was raised in a good (compared to bad) environment, participants inferred that they had greater free will and intent to commit the transgression, and evaluated them as having a more positive character. Environment did not significantly affect the Punishment decisions: $F(1, 132) = .513$, $p = .475$, $\eta^2_p = .004$.

There was a main effect of the type of Transgression (Harm vs. Purity) on evaluations of Character, $F(1, 132) = 19.599$, $p < .001$, $\eta^2_p = .129$, and Punishment, $F(1, 132) = 22.338$, $p < .001$, $\eta^2_p = .145$. For the harmful transgression, character evaluations were more negative and punishment decisions were harsher. The other main effects of Transgression were not significant; Intention: $F(1, 132) = .349$, $p = .556$, $\eta^2_p = .003$; Free Will: $F(1, 132) = .021$, $p = .885$, $\eta^2_p = .000$. Overall, the type of transgression committed influenced evaluations of character and deserved punishment, but did not influence evaluations of mental states.

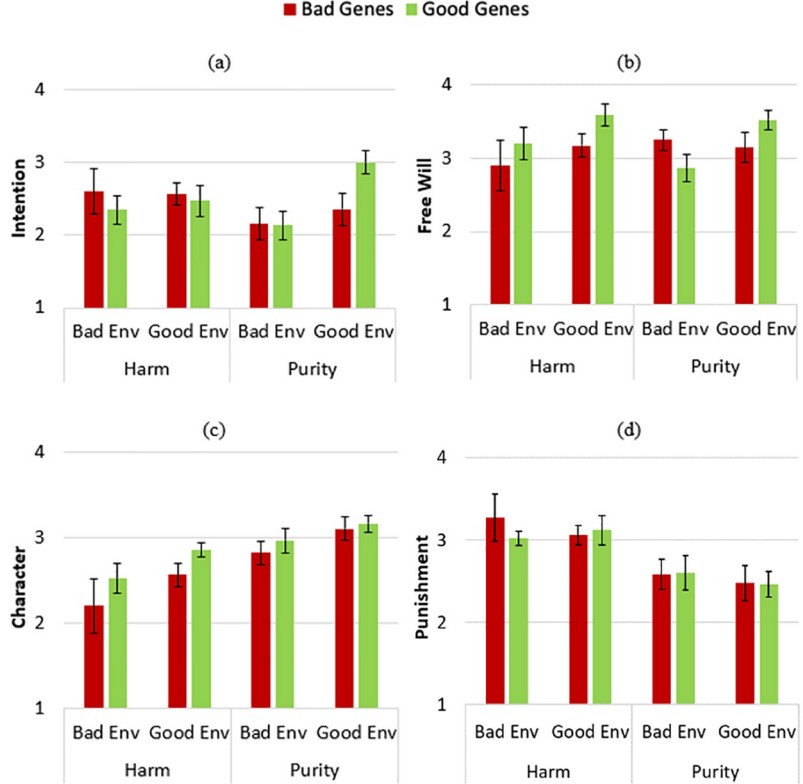

**Fig 4. Study 2 results.** Participants' average ratings (ranging from 1 to 4) of how much the perpetrator (a) desired to commit the transgression; (b) could have instead decided to act nicely; (c) had a positive character; (d) deserved punishment. Error bars represent standard errors of the mean.

There were no significant two-way or three-way interactions for any dependent variables, with all $p$s > .07. Thus, significant effects (and null effects) of Genes and Environment on evaluations of the perpetrator were similar for all dependent variables, whether the perpetrator had committed a moral violation or purity violation (see S1 File).

## Discussion

Similar to results from Study 1, in Study 2 we found that the quality of a perpetrator's genes and environment influenced judgments of their dispositional tendencies. When the perpetrator possessed bad (vs. good) genes, they were judged to have poorer character. When the perpetrator was raised in a bad (vs. good) environment, participants surmised that they had less free will. Whereas in Study 1 perpetrators with bad genes were judged to have greater intent, here, perpetrators who grew up in a positive environment were judged to have greater intent. Similar to Study 1, perpetrators who were raised in a positive environment were also judged to have greater free will in committing the transgression, as well as a greater desire to do so.

Yet, as in Study 1, the quality of the perpetrator's genes and environment had no influence on judgments of how much punishment they should receive. Punishment decisions were influenced only by the type of transgression (harm vs. purity violation); the perpetrator received greater punishment after committing the harm violation than after committing the purity violation. The type of transgression also influenced character judgments, such that the perpetrator who committed a harm violation was evaluated more negatively. The significant effect of

transgression type may be limited to the specific transgressions that we chose. To test the robustness of this effect, future research can use a broader range of transgressions, and ensure that harm and purity transgressions are matched on severity.

In Study 2, we found that the type of transgression influences punishment assessments, much more so than information about the perpetrator's background. This was true even for a transgression of moral purity that involved no physical harm and no specific victim. In Study 3, we attempt to divert participants' focus from the transgression even further, by turning to a case in which the perpetrator has no causal role in a negative outcome, by virtue of being lucky.

## Study 3

The wrongness of an action does not completely determine how that action will be punished. Individuals also take into account actors' intentions and the amount of harm caused by their actions [7]. In general, when there are no negative consequences of an action, there is no reason to punish the actor, as conveyed by the adage, "no harm, no foul". However, the phenomenon of "moral luck"–in which some negligent actors can happen to avoid harm by sheer chance, while others who act in similar ways can end up being much less fortunate—indicate that people's intuitions in these cases are not so clear-cut. When it is apparent that the punishment meted out for a harmful outcome is at least partially dependent on luck, it becomes prudent to begin assessing the relevant mental, dispositional, and situational characteristics that increased the likelihood of the outcome [51–52]. In line with this reasoning, we hypothesized that a norm violation which causes no direct harm could potentially lead to an increased focus on the genetic and environmental factors that led the perpetrator to become a particular "kind" of person, and thus influence punishment decisions. Specifically, we predicted that if two actors engaged in the same negligent action (reckless rock-throwing), resulting in a victim being injured, the morally lucky agent who threw rocks but did not directly cause the harm would receive some small amount of punishment that would depend in part on the actor's genetic and environmental background.

Similar to Studies 1 and 2, Study 3 investigated how information about genes and environments influence evaluations of intention, free will, character, and punishment. We included all of these dependent variables to keep our sequence of questioning similar to that of Studies 1 and 2 (so that we could neatly compare findings across studies), but our primary focus in this study was on participants' punishment decisions. In Study 3, we compared an *accidentally harmful* violation with a *morally luck*y situation that caused no direct harm. These scenarios were designed to investigate whether individuals' punishment decisions are influenced by an actor's genetic and environmental background in a case where there is the potential to cause harm, but no actual harm is caused (because of moral luck). In this study, the scenarios described the main character and his/her friend both throwing rocks off a bridge, such that they were equally negligent. In the Direct Harm condition, the main actor threw a rock which hit a person below. In the No Direct Harm scenario, the actor's *friend* threw the rock which hit the person below. Although both the main actor and the friend engaged in the same actions and had the same intentions, the cause of the outcome in each scenario differed. We were particularly interested in whether background information regarding genes and environment influenced how people made punishment decisions about the actor who was morally lucky.

## Methods

**Participants.**    Participants were recruited through Amazon Mechanical Turk and paid $0.80 each. They were again restricted to workers in the United States with a minimum HIT

approval rate of 95%. The survey was completed by 198 participants, but 50 (25%) were excluded for providing one or more inaccurate answers on comprehension check questions, leaving data from 148 participants (89 female, 1 non-binary, $M_{age}$ = 36.90, $SD_{age}$ = 11.90). A power analysis using G*Power [40] indicated that we required at least 128 participants to detect medium-sized main effects (e.g., $fs \geq$ .25), with statistical power = .80 and $\alpha$ = .05. Having 148 participants, our actual statistical power was .856.

**Design and procedure.** Participants completed the survey on Qualtrics. After providing consent and identifying their gender, each participant was randomly assigned to read a single gender-matched scenario in one of eight conditions, which varied by type of genes (Good or Bad), type of environment (Good or Bad), and type of outcome (Direct Harm or No Direct Harm). (The participant who did not report a binary gender received the scenario with a male character.) As in Studies 1 and 2, the switched-at-birth paradigm was used, with the possibility for being presented with information about a positive or negative genetic predisposition and a positive or negative rearing environment. In the Direct Harm condition, the vignette ended with a transgression involving throwing rocks from the top of a hill with a friend and accidentally hitting a person below, causing a severe injury. In the No Direct Harm condition, the main character was also throwing rocks, but his/her friend directly caused the severe injury to the person below (see Fig 5).

The participants were then asked seven questions, examining the same variables as in Studies 1 and 2; however some phrasing was changed to reflect the new transgression. Question 1 was: "Deep in her heart, did Jane/Fred want the rock to hit that person?" Question 2 was: "Could Jane/Fred have decided not to throw rocks and play nicely on the playground instead?" Question 3 was: "How much trouble should Jane/Fred get in for throwing rocks?" Questions 4–7 (punishment assessments, dangerousness, and goodness) remained the same as Studies 1 and 2. In the No Direct Harm condition, even though it was the friend of Jane or Fred who threw the rock that injured a person, participants were still asked about Jane's or Fred's role in the transgression (given that s/he was being equally negligent). All questions were recorded on the same 4-point Likert scales used in the previous two studies. Comprehension check questions remained the same as in Study 2, but also included one additional question to ensure that participants understood the details of the transgression: "Who hit the person with the rock?" As in the first two studies, we created a *Character* composite which included questions about the actor's Dangerousness (reverse coded) and Goodness ($\alpha$ = .704); and a *Punishment* composite which included the Trouble, Timeout, and Toys Confiscated questions ($\alpha$ = .755).

## Results

Utilizing a series of four 3-way ANOVAs, we tested for effects of information about the actor's Genes (Good vs. Bad), Environment (Good vs. Bad), and Outcome (Direct Harm vs. No Direct Harm) on evaluations of intention, free will, character, and deserved punishment (see Fig 6). We found a significant main effect of Genes for evaluations of Intention, $F(1, 140)$ = 12.799, $p <$ .001, $\eta^2_p$ = .084; Free Will, $F(1, 140)$ = 4.594, $p$ = .034, $\eta^2_p$ = .032; and Character, $F(1, 140)$ = 5.839, $p$ = .017, $\eta^2_p$ = .040. Participants who read about the actor's bad genes reported that they more strongly desired to commit the transgression, had less free will, and had worse character. There was no effect of Genes on ratings of Punishment, $F(1, 140)$ = 1.630, $p$ = .204, $\eta^2_p$ = .012.

There was a significant main effect of Environment on evaluations of Character, $F(1, 140)$ = 19.373, $p <$ .001, $\eta^2_p$ = .122; participants evaluated the actor more positively when they had a good (vs. bad) environmental background. There were no other significant main effects of Environment; Intention: $F(1, 140)$ = 1.324, $p$ = .252, $\eta^2_p$ = .009; Free Will: $F(1, 140)$ = .825, $p$ = .365, $\eta^2_p$ = .006; Punishment: $F(1, 140)$ = .173, $p$ = .678, $\eta^2_p$ = .001.

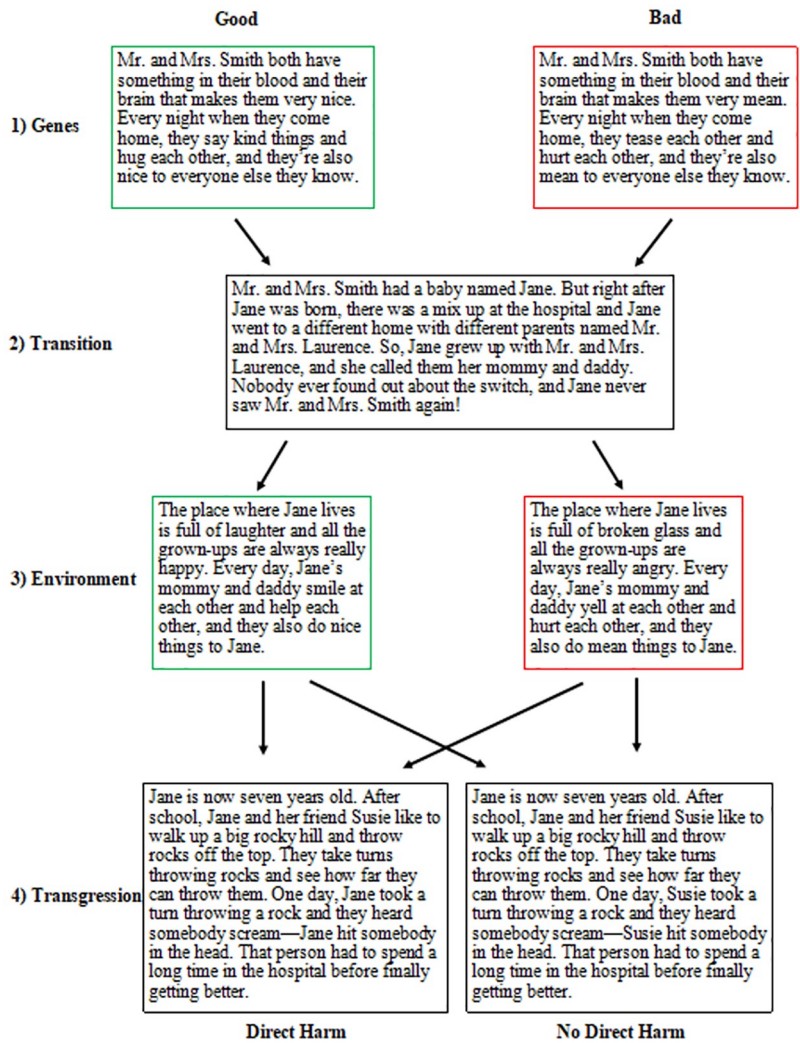

**Fig 5. Flow chart of Study 3 switched-at-birth vignettes and condition assignment.** Scenarios with good or bad genes were randomly paired with either good or bad environments and then randomly assigned to either the Direct Harm or No Direct Harm condition, resulting in a total of eight possible combinations. All participants received the same transition.

The only variable that influenced Punishment decisions was the type of Outcome (Direct Harm vs. No Direct Harm), $F(1, 140) = 7.657$, $p = .006$, $\eta^2_p = .052$. The amount of punishment assigned to the actor was greater when they accidentally caused harm than when the actor performed the same actions but did not directly cause harm (i.e., was morally lucky). There were no other significant effects of transgression type; Intention: $F(1, 140) = .260$, $p = .611$, $\eta^2_p = .002$; Free Will: $F(1, 140) = .125$, $p = .725$, $\eta^2_p = .001$; Character: $F(1, 140) = 1.732$, $p = .190$, $\eta^2_p = .012$. Thus, the type of transgression influenced punishment assessments, but not the other morally relevant assessments, such as the actor's character, intent, or level of control for committing the transgression.

There were no significant two-way or three-way interactions for any dependent variables (all $ps > .08$), suggesting that the effects of Genes, Environment, and Outcome did not moderate one another (see S1 File).

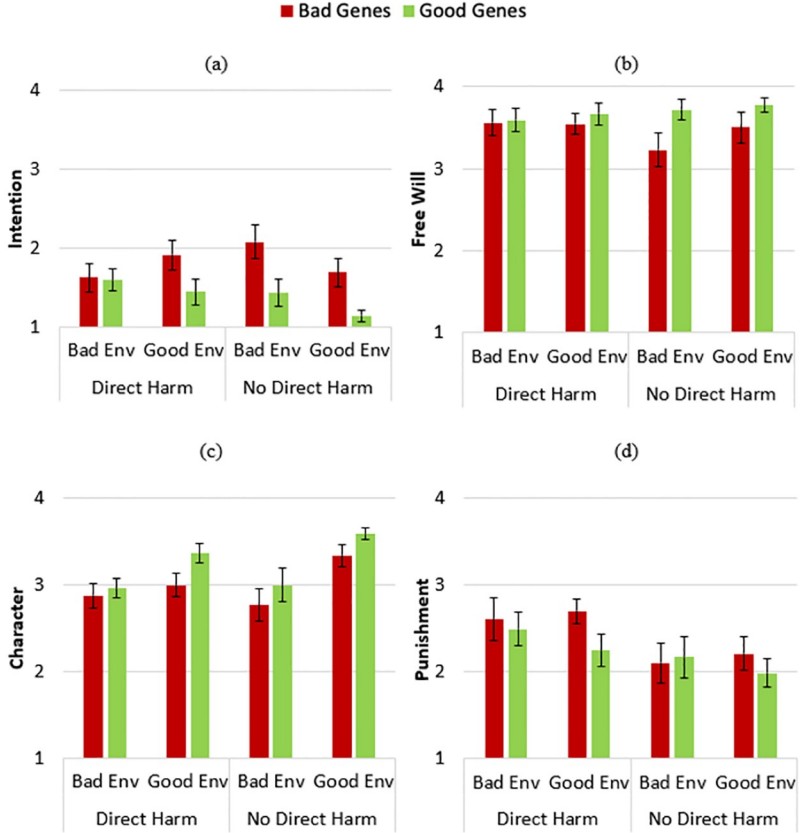

**Fig 6. Study 3 results.** Participants' average ratings (ranging from 1 to 4) of how much the perpetrator (a) desired to commit the transgression; (b) could have instead decided to act nicely; (c) had a positive character; (d) deserved punishment. Error bars represent standard errors of the mean.

## Discussion

Similar to Studies 1 and 2, the actor's genetic and environmental background influenced evaluations of their mental states and their character. When the actor possessed bad (vs. good) genes, participants inferred that the actor more strongly desired to commit the transgression, had less free will, and had poorer character. When the actor was raised in a bad (vs. good) environment, participants also inferred that they had a more negative character. Yet, the effects of information about the actor's genes and environment did not translate into effects on punishment assessments, replicating and extending our findings from Studies 1 and 2.

The only factor that did influence punishment assessments in Study 3 was the *outcome* of the transgression: despite the two actors behaving identically, greater punishment was delivered when the actor's behavior led to harm. This finding is similar to that of Study 2, in that the nature of the transgression influenced punishment assessments, whereas information about actor's genes or environment did not. This is consistent with previous research highlighting that the outcomes of violations tend to motivate punishment, even whether the actor does not intend for the outcome to occur [7, 53].

Unlike in Study 2, the type of transgression in Study 3 (throwing rocks and accidentally hitting a person, or throwing rocks and being morally lucky) did not affect evaluations of the actor's character. This suggests that participants interpreted the *action* (throwing rocks), rather than the particular outcome of the actions, as a better reflection of the actor's character. It is

possible that participants largely considered accidentally hurting someone and being lucky as uncontrollable factors, rather than outcomes related to the actor's dispositional tendencies.

## General discussion

Decades of psychological and biological research have revealed how people's background circumstances—both genetic and environmental—influence how they think, act, and the kind of people they become. Our studies highlight that associations between individuals' background circumstances and their psychological and behavioral outcomes are embedded in folk psychology; participants modulated their evaluations of a perpetrator's mental states and dispositions in accordance with information about the perpetrator's birth family and rearing environment. In Studies 1 and 3, the perpetrator's genetic background tended to be a particularly salient factor in participants' inferences about their intentions and free will, whereas in Study 2 the perpetrator's rearing environment tended to be a salient factor in participants' inferences about their free will. In all three studies, character assessments were especially malleable, influenced by information about both genetic and environmental backgrounds. Thus, while we cannot confidently conclude that perceptions of mental states are reliably influenced by information about nature or nurture, there is ample evidence to suggest that people rely on information about biological and environmental circumstances to make assessments of perpetrators' character traits.

Strikingly, despite the impact of genetic and environmental information on participants' inferences about internal characteristics of perpetrators, these factors were irrelevant in participants' evaluations of the amount of punishment that perpetrators deserved. Previous research has found that moral evaluations are directly influenced by information about actors' mental states [7, 53] and character [54–55]. Thus, there was reason to predict that background characteristics would affect punishment decisions, by virtue of the perpetrator's background influencing judgments of their intention, free will, and character. However, even though information about background characteristics reliably influenced participants' evaluations of the perpetrator's mental states and character across all three studies, this never translated into differences in punishment judgments. Instead, evaluations about the deservingness of punishment were based *only* on the nature of the crime; they were robustly immune to various extenuating circumstances.

One explanation for the null effect of background factors on punishment assessments is that punishment decisions (particularly as compared to assessments of moral wrongness) rely heavily on the *outcomes* of actors' behaviors [7]. Empirical and theoretical work has indicated that the signaling and pedagogical functions of punishment are most effective when punishment decisions are primarily driven by the presence of direct consequences, rather than by invisible factors such as a person's internal dispositions [8]. Previous research on extenuating circumstances has almost exclusively involved harmful actions [9, 24], and so it was reasonable to surmise that those outcomes were so egregiously negative that they might have overshadowed the relevance of background circumstances. However, we reduced the salience of outcomes in Studies 2 and 3, which involved an impure action or a negligent but inconsequential action, respectively, and punishment decisions remained unaffected by the actors' genetic and environmental histories. The current studies thus provide strong evidence that, across a variety of circumstances, background factors seem to be largely irrelevant in laypersons' punishment decisions, even in the absence of direct harm. This converges with findings from previous studies [56], in which the genetic backgrounds of perpetrators who commit everyday transgressions (e.g., stealing a cookie), have no influence on participants punishment judgments. Additionally, previous work has found that, although genetic information mitigated

evaluations of the perpetrator's control and increased beliefs about their reoffending, genetic information did not affect sentencing decisions [57].

However, the type of outcome in the current studies (i.e., a harm vs. a purity violation, or a directly harmful vs. morally lucky outcome) did influence the harshness of participants' punitive evaluations; thus, it is neither the case that participants ignored details of the transgressions nor that their punishment decisions were uniform and unmalleable. Similarly, information about genetic and environmental backgrounds influenced evaluations of mental states and character, and thus it is not the case that participants ignored the details of the actors' histories. Rather, these null effects are evidence against the relevance of information about perpetrators' genetic or environmental backgrounds in laypersons' punishment decisions.

Future research will need to further confirm these null effects—for instance, by testing whether they are due to nullifying competing effects across participants (with some participants treating genetic factors as mitigating and some participants treating genetic factors as aggravating) [24]. Although we did not find compelling evidence of genetic information acting as a "double edged sword", it will be important for future work to help reconcile our findings with some of the positive effects that have been previously uncovered by other researchers (e.g., [26–29]). For instance, future research could investigate whether participants' political ideologies moderate the effect of information about genetic or environmental factors on punishment decisions. If liberals and conservatives react to information about biological and environmental circumstances in opposing ways, these tendencies could negate each other when averaging across a full sample. In addition, future research could examine the extent to which individual differences in empathy or sympathy toward a perpetrator moderates the impact of information about that perpetrator's background.

Unlike most previous studies that have investigated severe transgressions such as murder and harsh sentences such as capital punishment [3, 19, 21], the current research focused on more everyday moral transgressions (e.g., throwing rocks on a playground) and more typical sanctions (e.g., time-out). Because judgments of transgressions are heavily influenced by the depravity or severity of the action, as well as their effects on victims (e.g., [58]), these judgments may also be differentially affected by a perpetrator's background characteristics. Thus, future research could use the current "switched-at-birth" paradigm to examine how adult participants may make judgments of punishment about transgressions of a range of severity, in order to see if the null effects observed in this research hold across such transgressions with a broad range of perceived depravity and negative consequences.

The vignettes used in the current studies were designed to be accessible to participants with varying educational backgrounds; for example, we did not include medical, biological, or sociological jargon to describe the perpetrators' biological and environmental circumstances. These same vignettes may be used with child participants, to examine how the inferences and judgments uncovered here tend to unfold across childhood. Although the language used in the vignettes was chosen to make the stimuli accessible to participants across a wide range of educational backgrounds, the generality of some language (e.g., referring to something in the perpetrators' "blood and brain" rather than specific genetic variants) could be a limitation. Extensions of the current work could include more precise language about perpetrators' genetic and environmental backgrounds. Additional work may also investigate whether the sequencing of information (e.g., whether extenuating circumstances are described before or after transgressions) alters punishment decisions, or whether perpetrators' backgrounds may matter more for punishment decisions regarding premeditated vs. spontaneous transgressions. Laying a foundation for this future work, the current studies identify a robust dissociation between information about perpetrators' histories and laypersons' punishment decisions. This

contradicts assumptions embedded within the U.S. legal system; for example, background character evidence is admissible during sentencing phases in certain cases, yet the current studies (and prior work) suggest that such evidence may actually have little bearing on sentences [59]. It is incumbent on researchers to identify ways in which such evidence can be presented such that it meaningfully shapes punishment decisions.

## Supporting information

**S1 File. Supplementary materials.**
(DOCX)

## Acknowledgments

We are grateful to Emily Conder, Christopher Jaeger, Xinjie Zhao, and Valerie Zizik for their assistance in the initial conceptualization of the research and to Lulu Gomez, Nicole Kolmstetter, and Jennifer Shabrach for their assistance with coding open-ended responses.

## Author Contributions

**Conceptualization:** Julianna M. Lynch, Jonathan D. Lane, Joshua Rottman.

**Formal analysis:** Julianna M. Lynch, Joshua Rottman.

**Funding acquisition:** Joshua Rottman.

**Investigation:** Julianna M. Lynch, Jonathan D. Lane, Colleen M. Berryessa, Joshua Rottman.

**Methodology:** Julianna M. Lynch, Jonathan D. Lane, Colleen M. Berryessa, Joshua Rottman.

**Supervision:** Joshua Rottman.

**Writing – original draft:** Julianna M. Lynch.

**Writing – review & editing:** Jonathan D. Lane, Colleen M. Berryessa, Joshua Rottman.

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
