## [Decision Letter · Decision Letter 0]

26 Jul 2019

PONE-D-19-15890

How Information about Perpetrators’ Nature and Nurture Influences Assessments of their Character, Mental States, and Deserved Punishment

PLOS ONE

Dear Dr. Rottman,

Thank you for submitting your manuscript to PLOS ONE. After careful consideration, we feel that it has merit but does not fully meet PLOS ONE’s publication criteria as it currently stands. Therefore, we invite you to submit a revised version of the manuscript that addresses the points raised during the review process.

Please find the reviewers' comments, as well as mine, below.

We would appreciate receiving your revised manuscript by Sep 09 2019 11:59PM. To enhance the reproducibility of your results, we recommend that if applicable you deposit your laboratory protocols in protocols.io, where a protocol can be assigned its own identifier (DOI) such that it can be cited independently in the future. For instructions see: http://journals.plos.org/plosone/s/submission-guidelines#loc-laboratory-protocols

We look forward to receiving your revised manuscript.

Kind regards,

Valerio Capraro

Academic Editor

PLOS ONE

**Journal Requirements:**

3.Please include captions for your Supporting Information files at the end of your manuscript, and update any in-text citations to match accordingly. Please see our Supporting Information guidelines for more information: http://journals.plos.org/plosone/s/supporting-information.

**Additional Editor Comments (if provided):**

I have now collected four reviews from four experts in the field. Although all the reviewers agree that the topic of the paper is important, their overall judgment about the paper is split: one recommends minor revision, two recommend major revision, one recommend rejection. In light of this, I would like to invite you to revise your paper following the reviewers' suggestions. Needless to say that all comments must be addressed. However, particular care should be given to the comments from the negative reviewer (Reviewer 2). Her/his main comment is that three studies are underpowered. The reviewer explicitly suggests to re-run the studies with a larger sample. In your revision, you should, at the very least, convince the reviewer (and the general reader) that the small samples are not a major issue in your context (note that this is a very important point, especially in light of the current Replicability Crisis). Of course, if you have the possibility to rerun the studies with larger samples, this would certainly be much better.

Reviewers' comments:

Reviewer's Responses to Questions

**Comments to the Author**

1. Is the manuscript technically sound, and do the data support the conclusions?

Reviewer #1: Yes

Reviewer #2: No

Reviewer #3: Yes

Reviewer #4: Partly

2. Has the statistical analysis been performed appropriately and rigorously? 

Reviewer #1: Yes

Reviewer #2: No

Reviewer #3: Yes

Reviewer #4: Yes

3. Have the authors made all data underlying the findings in their manuscript fully available?

Reviewer #1: Yes

Reviewer #2: Yes

Reviewer #3: Yes

Reviewer #4: Yes

4. Is the manuscript presented in an intelligible fashion and written in standard English?

Reviewer #1: Yes

Reviewer #2: Yes

Reviewer #3: Yes

Reviewer #4: Yes

5. Review Comments to the Author

Reviewer #1: The authors used a switched-at-birth paradigm to examine how information about nature and nurture affects inferences about perpetrators, as well as recommended punishments. The results indicate that this information affected judgments about the perpetrator, but not the recommended punishment. The paper is well-written and presents some interesting findings.

I suggest making some minor changes to the method sections. In Study 1 it would be helpful to first introduce the dependent measures in the context of explaining the proposed composite measures. A brief overview of the phases of the procedure should be provided in the text along with the flowchart so that readers will be able to understand what “the same transition” refers to without having to look at the chart. Also, the way the participants section is written in Study 1 implies that “adult participants” will be followed by a description of child participants.

Punishment is defined as attempting to alter or reform the bad behavior, either socially or legally. However, it can also send messages that have nothing to do with the person who committed the transgression, either as a signal to others who might be thinking of committing a crime, or as a signal to victims that their suffering matters (which can help to explain why outcomes matter so much).

In the description of Study 2, it is unclear whether people would accept that genetic influences relating to being nice or mean could have implications for running around on the playground naked and shouting dirty words. Also, in Study 2, I wondered whether the difference in punishment recommendations between the harm and purity transgression types is really about harm, or about how severe the transgressions are perceived to be. What if you replaced the harm transgression with one in which the level of harm was likely to be small, such as hurting someone’s feelings by calling him stupid, or spilling someone’s juice? It might also matter if the harm were incidental to some other activity.

The authors note that the current research focuses on everyday transgressions rather than the kinds of severe ones that are typically studied. It would be helpful to explicitly lay out why these contexts might lead to different findings, and to describe how the current results map onto what has been observed for more serious transgressions. I also wonder how adults would reason about more serious transgressions using the current paradigm.

It is interesting that people can use information about genes to help them assess a range of characteristics (like dangerousness, attributions of desire to commit immoral acts, and perceptions of controllability) but not punishment. I wonder whether the lack of effects hold when looking only at the more general version of the punishment question (“How much trouble should Jane/Fred get into for hurting the other children?”).

Outcomes are important for punishment recommendations, as is suggested, but there are other possibilities too. It could be that genetic factors have effects that go in opposite directions, as might be the case if attributions of desire to commit immoral acts alone would lead to increases in punishment recommendations, and decreases in perceptions of controllability alone would lead to decreases in punishment recommendations. More explanation is also needed regarding how the lack of punishment effects can be understood in light of the evidence mentioned in the introduction that characteristics that are seen as linked to genetic influences can mitigate punishment assessments.

Reviewer #2: The authors neglected a fairly large corpus of research conducted by Appelbaum and colleagues that has examined this exact issue. In particular, his 2016 piece in the Journal of Law and Biosciences examined the impact of biological influences on misbehavior in non-criminal settings and in children. Some discussion of that study (and the others conducted by his group) is warranted. There are also numerous studies published in basic social psychology journals that examine how genetics affect attributions made by laypeople.

Study 1 had 9 cells and 201 participants. That amounts to 22 participants per cell. On its face, that is drastically underpowered. Especially in light of the fact that the previous literature finds null or small effects, the sample size must be large to be able to detect an effect assuming it does exist. (It also makes observed “statistically significant” effects unlikely to be true effects; see Button et al Nature Neuroscience for an explanation of this principle.) This sample size is inadequate to study this phenomenon. The problem is even more dramatic in the subsequent studies.

The studies should be re-ran with 5 to 6 times the amount of participants, at least. If the findings replicate, then I would recommend publication.

Reviewer #3: This is a well written paper that uses a creative approach (the switched at birth paradigm) to attempt to disentangle and assess the effects of information about "nature" and "nurture" on judgments of a perpetrator of wrongdoing. I would summarize the authors' findings as follows: when someone is harmed (studies 1 and 2), information about the wrongdoer's "bad genes" leads people to perceive her/him as having less free will, more intentionality, and worse character, while information about the wrongdoer's "bad environment" only seem to affect judgments of character (negatively). By contrast, in Study 2, the authors found no main effects of "bad genes" (though they found one marginal effect), while "bad environments" led to lower ratings of free will, lower ratings of intentionality, and worse character ratings. I found it striking that the results of studies 1 and 3 were so similar, while the pattern of results in study 2 was quite different. However, rather than highlighting the different pattern of results for study 2, the authors seem to saw in the first sentence of study 2's discussion section that the results were similar, which I found surprising. It might be more effective for the authors to explore/explain why study 2 might have had such a different pattern of results. It's interesting that in all of the studies, judgments of intentionality and free will were significantly affected by either genetic or environmental explanations, but seemingly never by both, whereas character ratings were seemingly affected by both kinds of explanations. The authors may wish to explore/discuss why this may be. Also interesting is the finding that punishment judgments, while sensitive to the amount of harm caused by the wrongdoing, do not seem to be responsive to genetic or environmental explanations. The lack of an effect on punishment is similar to what others have found (see the work of Paul Appelbaum and the work of Steve Heine). Overall, this is an interesting paper, though it might be useful to frame the findings a bit differently and explore why the patterns of results may have emerged the way they did.

Reviewer #4: Summary: Across three studies, adult participants were asked to evaluate a child perpetrator’s 1) intention, 2) free will, 3) character, and 4) appropriate punishment. In all three studies, the perpetrator’s backgrounds were manipulated with a “switched-at-birth” paradigm, presenting participants with scenarios featuring good or bad rearing environments and good or bad (implied) biological/genetic risk factors (i.e., parents with something bad or good in the blood and brain that made them mean or nice).

-Study 1: A perpetrator was described as interpersonally harmful (throwing rocks). If this individual had “bad genes,” he/she was perceived as having reduced free will (for perpetrators raised in neutral or good environments), increased intention, and worse character. There was also a single main effect for environment, with perpetrators perceived as having better character if raised in a neutral vs. bad environment.

-Study 2: Perpetrators were described as either interpersonally harmful (throwing rocks) or purity norm violating (running naked and swearing). Main effects here were mostly for environment, on intention (good envt. = greater intent), free will (good envt. = greater free will), and character (good envt. = better character). (“Bad genes” yielded marginally worse character judgments.) Participants also punished the harmful act more than the purity norm violation. (Note: Figure 4 suggests the opposite; I’m assuming the mistake is in the figure, as the text reports this same transgression main effect later in the Discussion.)

-Study 3: Perpetrators were described as either causing harm accidentally (by throwing rocks and hitting someone below) or lucky (having a friend who harms instead, and the actor’s rock misses). Perpetrators with bad genes (vs. good) were seen to have more intent, less free will, and worse character. Perpetrators raised in a good environment (vs. bad) were seen to have better character. The only effect on punishment was a main effect for transgression; individuals who had not inflicted direct harm received lower punishment ratings.

Review: Authors engage with a literature featuring mixed findings regarding how a perpetrator’s genetic vs. environmental background modifies people’s judgments. Across three studies, they demonstrate that backgrounds matter for certain judgments (free will, intention, character) but not for punishment.

I liked many aspects of this paper. Authors do a good job of concisely summarizing relevant literature. Methodologically, they do a good job in investigating various transgression types to strengthen their argument that backgrounds are largely irrelevant in punishment decisions. Specifically, after establishing that participants do not factor in background in punishing standard harms (Study 1), they move on to purity violations (Study 2) and moral luck scenarios (Study 3), for which prior literature hints that participants will be more focused on character/instrinsic factors, and less focused on the harm itself. Despite these changes, however, backgrounds did not matter for punishment decisions. I thought the variation in transgression type was a quite clever manipulation and made good use of prior findings from the broader literature, and the studies flowed well one to the other.

There were two elements of the paper that I found lacking. I elaborate below:

1) The design (as authors acknowledge) is targeted to a developmental population. Authors argue that the design works equally well for adults vs. children, but I do not agree. For one thing, the perpetrator is a child, which for an adult study is an odd choice. It is not clearly wrong, however; I think one could make the argument that choices about punishment for a child vs. adult are still interesting and relevant. But there is another larger problem here: The genetic background manipulation relies entirely on inference. Participants are not told at all about bad or good genes; instead they are told about “something in the blood and brain” that makes the biological parents mean or nice. Participants then must infer that the meanness or niceness is genetic, and that the “bad” or “good” genes are heritable. Though this phrasing makes sense if there were child subjects as part of the project, for a fully adult study, it’s just not a sensible phrasing. Authors argue that this simplified language is appropriate, but again I do not agree, or at least I’m not sure without a manipulation check. Specifically, do adults interpret these scenarios as involving genes? Or, if the authors wanted to develop a manipulation check that could be used with children, do participants interpret these scenarios as involving a feature that passes down from parent to child?

2) Findings are not consistent across the three studies; effects for genes and environment are not reliably present. (Genes had effects for Studies 1 and 3 and environment mostly did not; environment had effects for Study 2 and genes mostly did not.) I don’t want to dwell too long on this, as this is not the fault of the authors and was not a flaw of the studies (though I suppose it might be partly a result of a weak or variably interpreted manipulation, see above). Ultimately, we just can’t conclude anything unambiguously from the mixed findings that emerged. As well, authors face the challenge of presenting a null finding as one of the most important results of the studies, which is just inherently a hard thing to do. This probably weakens the overall appeal of the paper. Again, though, I see this as a minimal weakness.

Ultimately I would still want to see this published with some kind of assurance about the manipulation described in 1) above. Broadly speaking I also look forward to seeing this project come to fruition with children, and/or replicated/extended with a more appropriate adult design (e.g., with an adult perpetrator, and using more elaborated and adult-appropriate background manipulations), but of course in another paper.

Other notes:

-If space permits, I would like to see the choice options (or the anchors) for the Likert scales included somewhere in the paper, rather than in supplementary materials.

-Was gender included as a factor in initial analyses? If gender was not significant, stating this would be helpful.

-As noted in my summary, I think there is a mistake in Figure 4, for the Punishment variable. The text describes punishment as being harsher for harm vs. purity violations in Study 2, which is the opposite pattern from that depicted in Figure 4.

-Could authors clarify in the text what exactly was being tested in the essentialism measures they used? (What kind of essentialism was being measured? Global? Essentialism of aggression/criminality/prosociality? Some other kind of social essentialism?)

END REVIEW

6. PLOS authors have the option to publish the peer review history of their article (what does this mean?). If published, this will include your full peer review and any attached files.

Reviewer #1: No

Reviewer #2: No

Reviewer #3: No

Reviewer #4: No

---

## [Author Response · Author response to Decision Letter 0]

6 Sep 2019

We have uploaded a cover letter and response to reviewers, in which we respond to each of these comments in turn.

---

## [Decision Letter · Decision Letter 1]

30 Sep 2019

PONE-D-19-15890R1

How Information about Perpetrators’ Nature and Nurture Influences Assessments of their Character, Mental States, and Deserved Punishment

PLOS ONE

Dear Dr. Rottman,

Thank you for submitting your manuscript to PLOS ONE. After careful consideration, we feel that it has merit but does not fully meet PLOS ONE’s publication criteria as it currently stands. Therefore, we invite you to submit a revised version of the manuscript that addresses the points raised during the review process.

Please find below the reviewers' comments.

We would appreciate receiving your revised manuscript by Nov 14 2019 11:59PM. To enhance the reproducibility of your results, we recommend that if applicable you deposit your laboratory protocols in protocols.io, where a protocol can be assigned its own identifier (DOI) such that it can be cited independently in the future. For instructions see: http://journals.plos.org/plosone/s/submission-guidelines#loc-laboratory-protocols

We look forward to receiving your revised manuscript.

Kind regards,

Valerio Capraro

Academic Editor

PLOS ONE

Journal Requirements:

Additional Editor Comments (if provided):

The reviewers are pretty happy with you revision, but especially one of them suggests some more minor revisions before publication. Please address these last comments at your earliest convenience. I am looking forward for the revision.

Reviewers' comments:

Reviewer's Responses to Questions

**Comments to the Author**

1. If the authors have adequately addressed your comments raised in a previous round of review and you feel that this manuscript is now acceptable for publication, you may indicate that here to bypass the “Comments to the Author” section, enter your conflict of interest statement in the “Confidential to Editor” section, and submit your "Accept" recommendation.

Reviewer #1: All comments have been addressed

Reviewer #2: All comments have been addressed

Reviewer #3: All comments have been addressed

Reviewer #4: (No Response)

2. Is the manuscript technically sound, and do the data support the conclusions?

Reviewer #1: Yes

Reviewer #2: Yes

Reviewer #3: Yes

Reviewer #4: Yes

3. Has the statistical analysis been performed appropriately and rigorously? 

Reviewer #1: Yes

Reviewer #2: Yes

Reviewer #3: Yes

Reviewer #4: I Don't Know

4. Have the authors made all data underlying the findings in their manuscript fully available?

Reviewer #1: Yes

Reviewer #2: Yes

Reviewer #3: Yes

Reviewer #4: Yes

5. Is the manuscript presented in an intelligible fashion and written in standard English?

Reviewer #1: Yes

Reviewer #2: Yes

Reviewer #3: Yes

Reviewer #4: Yes

6. Review Comments to the Author

Reviewer #1: I am satisfied with this revision. While I understand there are good reasons to not include analyses of the explanation data that was collected in Studies 2 and 3, I think it will be important in future research to look closely at this kind of data to understand why information about bad genes and bad environments does not seem to affect punishment decisions. One possibility is that there are different effects going in different directions, for example if people feel sympathy for individuals who face challenges in life due to no fault of their own but also see them as likely to cause future harm. The former could lead to recommending of reduced punishment and the latter could lead to recommending increased punishment. I also wonder whether people’s political views might play a role in their ratings and justifications. This possibility is worth mentioning in the section that considers possible individual differences.

Reviewer #2: (No Response)

Reviewer #3: I have no objections to the changes that the authors have made in response to the comments I provided on their original submission.

Reviewer #4: I believe that authors appropriately handled my concerns except for the issue of the requested manipulation check regarding Ss’ interpretation of the genetic backgrounds. (In brief, my original concern was with authors’ decision to instantiate the “bad” or “good” genetic conditions by providing information about substances in the blood or brain (e.g., a participant in the good “genetic” background condition would read about “something in the blood and brain [of the parents] that made them very nice.”). Authors argue that their open-ended coding of explanations, now described in a footnote, reveals that genetic interpretations were common, as a sizable number of participants mentioned DNA, genetics, or inheritance. Though somewhat reassuring, this does not tell us about the success of the genetic manipulation in direct/absolute terms (i.e., what percentage of Ss interpreted that description as genetic, vs. some other cause like disease, drugs, or poisoning?) or in relative terms (are the environmental vs. genetic interpretations made with equal or similar levels of success?). Of course after the fact, these data cannot be obtained, but a post-hoc examination of a separate sample could have supplied considerably more relevant data to answer these questions.

I would have preferred to see such a manipulation check, but I don’t want that preference to be obstructive in moving this project toward publication. The (maybe) easier fix that I would then request is to do the following: 1) Briefly mention, in a couple sentences, the explicit choice of “blood and brain” language in the Method or the lead-up near the end of the Introduction (rather than referring the reader to a figure). (This would also have the added benefit of making the current footnote more comprehensible, as it would set up a rationale for the open-ended coding described therein), and 2) Briefly mention, in the Discussion, extensions or variations of the study involving more direct language that conveys the crucial heritable feature of the “genes” backgrounds. Again, I completely understand why the chosen language was used and see that it features a number of benefits, but I also think potential drawbacks should be acknowledged somewhere.

Absent that, run a quick sample of MTurkers and show that a genetic (and crucially inheritance-focused) interpretation is made by the majority of the respondents, and at levels comparable to people’s interpretation of the upbringings as environmental.

7. PLOS authors have the option to publish the peer review history of their article (what does this mean?). If published, this will include your full peer review and any attached files.

Reviewer #1: No

Reviewer #2: No

Reviewer #3: No

Reviewer #4: No

---

## [Author Response · Author response to Decision Letter 1]

3 Oct 2019

We have included a response to the remaining reviewers' comments in our Cover Letter and Response to Reviewers file.

---

## [Editor Report · Decision Letter 2]

7 Oct 2019

How Information about Perpetrators’ Nature and Nurture Influences Assessments of their Character, Mental States, and Deserved Punishment

PONE-D-19-15890R2

Dear Dr. Rottman,

We are pleased to inform you that your manuscript has been judged scientifically suitable for publication and will be formally accepted for publication once it complies with all outstanding technical requirements.

With kind regards,

Valerio Capraro

Academic Editor

PLOS ONE
---

## [Editor Report · Acceptance letter]

14 Oct 2019

PONE-D-19-15890R2 

How Information about Perpetrators’ Nature and Nurture Influences Assessments of their Character, Mental States, and Deserved Punishment 

Dear Dr. Rottman:

I am pleased to inform you that your manuscript has been deemed suitable for publication in PLOS ONE. Congratulations! Your manuscript is now with our production department. 

With kind regards,

on behalf of

Dr. Valerio Capraro 

Academic Editor

PLOS ONE